# OpenReview forum: "Value Drifts: Tracing Value Alignment During LLM Post-Training"
_ICLR.cc/2026/Conference — Submitted to ICLR 2026_

### Official Review · Reviewer_9xf6 · 2025-10-20

**Soundness:** 3
**Presentation:** 3
**Contribution:** 2
**Rating:** 2
**Confidence:** 5

**Summary:**

This paper traces value alignment throughout different parts of an LLM’s lifecycle. From pretraining, to supervised fine-tuning, to post-training. First, they measure value drift from base models and after SFT, and show how this method shifts values significantly. Then, they explore the effect of PPO, DPO, and SimPO, and see very little or no value drift. Finally, they construct a dataset that either supports or opposes values, increasing the value gap, and find that DPO drifts considerably as a response.

**Strengths:**

- The paper covers different datasets, methods, and models.
- It’s relevant to study what happens to the underlying values of these widespread models during different training phases.
- The writing of the paper is very clear.
- There are a lot of details in the paper and appendix, which highlight its reproducibility.

**Weaknesses:**

I think studying the value drifts of these models at different moments of the training is important. However, in this paper, you approach this problem mainly from the perspective of the methods used for value alignment. To me, what matters most is the interaction with the data, and you only cover a bit of that at the end in Section 6.

In Section 4, you show the effects of SFT with two different datasets and models. Using Wildchat and Alpaca leads to different values, but you don’t explore the prior distribution of these datasets, even though you mention that as a possible explanation. Aren’t you just showing that SFT works? That doesn’t seem surprising to me, and you also acknowledge that this is known (line 300).

In Section 5, you show the value drifts after SFT with three algorithms (DPO, PPO, SimPO) and show that there are no significant shifts. Once again, you don’t show the prior distribution of the datasets used, even though later you mention this might be an issue.

Finally, Section 6 deals with data biases and the effects of the above-mentioned algorithms. You find very little change from PPO, but as you say, there’s a penalty for separating from the SFT. SimPO suffers from no shifts as well, but DPO shifts values considerably. Once again, I feel like you’re just showing whether these algorithms work or not, and that without saying much about the data doesn’t show anything that we shouldn’t already know.

From my perspective, it’d be better to study this problem mostly from a data x algorithm point of view. For example, you could generate different datasets with different distributions, such as bimodal, and see how those affect post-training. Do models just align with the mode that skews it more? Does it result in a neutral-value model? Does it overfit to support or oppose with each specific mode? I see a lot more open questions in this approach.

**Questions:**

- You mention 4 models (Llama3 3B and 8B + Qwen3 4B and 8B) but show results only for the 3B and 4B models. Am I missing something? I see Tables 10 and 12 in the Appendix, but those are just SFT results.

---

> ### Author Response · Authors · 2025-11-26
> **Response to Reviewer 9xf6 (1/2)**
>
> We thank the reviewer for noticing our work as a relevant and important study and appreciate the reviewer's comments about the writing and the details of the experiments.
>
> **W1:** *However, in this paper, you approach this problem mainly from the perspective of the methods used for value alignment. To me, what matters most is the interaction with the data, and you only cover a bit of that at the end in Section 6.*
>
> **Response:** As mentioned on line 047, our main research question is to trace value alignment during post-training by understanding how the underlying training data, algorithms, and their interaction shape the values expressed by a model. While the role of the underlying data is crucial, a full audit is beyond the scope of this work, so we can only approximate the dataset distributions to provide insights into their influence, **as described below for W2**
>
> ---
>
> **W2.** *In Section 4, you show the effects of SFT with two different datasets and models. Using Wildchat and Alpaca leads to different values, but you don’t explore the prior distribution of these datasets, even though you mention that as a possible explanation. Aren’t you just showing that SFT works? That doesn’t seem surprising to me, and you also acknowledge that this is known (line 300).*
>
> **Response:** To approximate the stance distribution of SFT datasets, we use two methods:
>
> *Approach 1:* Embedding-based retrieval. We embed each SFT dataset (WildChat and Alpaca) and all value-probing questions in V-PRISM using the all-mpnet-base-v2 sentence transformer. For every question in V-PRISM, we compute cosine similarity against all dataset embeddings and retain datapoints with similarity ≥ 0.5. We then classify the stance for each retrieved datapoint, following the same evaluation protocol used for model outputs (§3). Using this method, we retrieve 5996 datapoints from WildChat (≈2.3%) and 612 datapoints from Alpaca (≈1.3%).
>
> *Approach 2:* Keyword-based retrieval. To capture a broader sense of the data distribution, we use a keyword-based search aligned with the 11 topics in V-PRISM. This way, we retrieve 37856 datapoints from WildChat dataset (≈14.8%) and 2110 datapoints from Alpaca dataset (≈4.5%), and apply our stance classification pipeline to these datapoints.
>
> We then plot these stance distributions, as shown in Appendix K1 in the revised manuscript. We observe that WildChat exhibits a predominantly neutral stance, with 72.3% of its retrieved data points classified as neutral. On the other hand, Alpaca exhibits a clear supportive stance, with a majority (67%) of its datapoints classified as supportive across all topics, which supports the point on line 300, that Alpaca datasets contain an implicit bias towards over-agreeable (supportive) responses.
>
> Our findings not only reflect that SFT works, but also reveal how **different instruction-following datasets impart different value** profiles, even though neither is designed explicitly for value learning.
>
> ---
>
> **W3.** *In Section 5, you show the value drifts after SFT with three algorithms (DPO, PPO, SimPO) and show that there are no significant shifts. Once again, you don’t show the prior distribution of the datasets used, even though later you mention this might be an issue.  Finally, Section 6 deals with data biases and the effects of the above-mentioned algorithms. You find very little change from PPO, but as you say, there’s a penalty for separating from the SFT. SimPO suffers from no shifts as well, but DPO shifts values considerably. Once again, I feel like you’re just showing whether these algorithms work or not, and that without saying much about the data doesn’t show anything that we shouldn’t already know.*
>
> **Response:** Similarly to the methodology described in W2 for SFT datasets, for standard preference datasets, we retrieve 10177 datapoints from UltraFeedback (16.7%) and 7627 datapoints from HH-RLHF (12.5%). We then calculate and plot the Euclidean distances between the stance distribution of the ‘chosen (preferred)’ and ‘rejected (dispreferred)’ responses. As shown in Figure 16 (see Appendix K2 in revised manuscript), we observe that both datasets are highly concentrated at the low end of the distance scale, exhibiting a large peak near 0. This indicates a low value gap and weak signal for preference optimization.
>
> In contrast to standard preference datasets, our synthetic dataset with a controlled value gap, the Euclidean distances show a dominant peak at the high-distance end, demonstrating a substantial number of preference pairs with a strong value difference, providing a clearer signal for preference learning (see Appendix K3 in the revised manuscript).

---

> > ### Author Response · Authors · 2025-11-27
> > **Response to Reviewer 9xf6 (2/2)**
> >
> > **W4:** *From my perspective, it’d be better to study this problem mostly from a data x algorithm point of view. For example, you could generate different datasets with different distributions, such as bimodal, and see how those affect post-training. Do models just align with the mode that skews it more? Does it result in a neutral-value model? Does it overfit to support or oppose with each specific mode? I see a lot more open questions in this approach.*
> >
> > As we claim in the paper on L47, our main research question is to study value alignment by disentangling the effects of datasets vs algorithms and their interaction. However, we thank the reviewer for the experiment suggestion. To directly study this, we extend the setup from Section 6 and use the synthetic preference dataset to construct three controlled preference datasets with different stance distributions over the same prompts:
> >
> > **[Setting 1] A balanced bimodal preference dataset** (i.e., 50% support and 50% oppose as chosen preference across all topics)
> >
> > **[Setting 2] A support-dominant preference dataset** (i.e., 70% support and 30% oppose as chosen preference across all topics)
> >
> > **[Setting 3] An oppose-dominant preference dataset** (i.e., 30% support and 70% oppose as chosen preference across all topics)
> >
> > We use Llama3-3B as the base model and run all preference optimization algorithms, initializing SFT with WildChat and SFT with Alpaca. We report full results in Appendix O in the revised manuscript.
> >
> > First, we observe that across all settings and topics, PPO and SIMPO exhibit minimal to no value drifts. For DPO, the answer is nuanced. In Setting 1, we do not end up with a neutral-value model. The policy pushes both preferences (support and oppose), but the updates are anchored with respect to the SFT prior. For example, for the topic of Immigration, we observe that the SFT model has a dominant neutral stance. However, after DPO training, the probability of neutral stance decreases, and both support and oppose stance probabilities increase, with the support stance being the final dominant stance. We refer the reviewer to Figure 21 in Appendix O of the revised manuscript.
> >
> > Next, in Settings 2 and 3, when support or oppose is the dominant preference, we observe that the model does drift towards the chosen preference. But, since the updates are anchored with respect to the SFT prior, it might not become the dominant preference. We refer the reviewer to Figure 22 in Appendix O for examples.
> >
> > ---
> >
> > **Q1:** *You mention 4 models (Llama3 3B and 8B + Qwen3 4B and 8B) but show results only for the 3B and 4B models. Am I missing something? I see Tables 10 and 12 in the Appendix, but those are just SFT results.*
> >
> > **Response:** Thank you for pointing this out. We include the results for Llama3 8B and Qwen3 8B in Appendix P in the revised manuscript.

---

> > > ### Comment · Reviewer_9xf6 · 2025-11-28
> > >
> > > > **W1** As mentioned on line 047, our main research question is to trace value alignment during post-training by understanding how the underlying training data, algorithms, and their interaction shape the values expressed by a model. While the role of the underlying data is crucial, a full audit is beyond the scope of this work, so we can only approximate the dataset distributions to provide insights into their influence, as described below for W2
> > >
> > > I can see you emphasized alignment during post-training. I think the disconnect is that I (and other reviewers) believe that this approach is limited to verifying how these well-characterized algorithms work. A full audit of the data is not necessary, more on that below.
> > >
> > > > **W2** Our findings not only reflect that SFT works, but also reveal how different instruction-following datasets impart different value profiles, even though neither is designed explicitly for value learning.
> > >
> > > Thank you for taking the time to analyze the data, it's very useful to see these results. To me, that confirms that SFT works and transfers the underlying values, which is expected.
> > >
> > > > **W3** This indicates a low value gap and weak signal for preference optimization.
> > >
> > > Once again, I think this shows why some methods don't work well, and justifies your use of synthetic data. You might disagree, but so far this isn't a significant contribution.
> > >
> > > > **W4** we extend the setup from Section 6 and use the synthetic preference dataset to construct three controlled preference datasets with different stance distributions over the same prompts
> > >
> > > Thank you for considering my suggestion, I personally think this is a really interesting direction. This is promising, but the paper would have to focus on this direction, explore in depth, and reveal interesting insights about the post-training methods to be a significant contribution for publication at a top conference.
> > >
> > > ### Minor
> > >
> > > There's a typo in line 2063 "chosne"

---

### Official Review · Reviewer_L3se · 2025-10-31

**Soundness:** 3
**Presentation:** 3
**Contribution:** 2
**Rating:** 4
**Confidence:** 4

**Summary:**

In the paper, the authors focused on an important topic, how models acquired value preference through post-training. In particular, they proposed a method to quantify the shift of model value during post-training phases: 1) SFT and 2) RL. They empirically exam this proposed method for 4 models on 4 datasets against those 2 post-training phases and presented their findings.

**Strengths:**

The paper is well-written and easy to follow and understand. And the authors studied a very interesting and important topic: the models’ output value preference and how the models acquired the value preference through post-training.

**Weaknesses:**

I listed a few weaknesses as below:
1. Overall, the paper is more like an empirical study of model value alignment. It did not propose novel algorithm or framework to perform better model human value alignment, neither construct new dataset or benchmark to perform model human value evaluation. The contribution of the paper is somewhat limited. Seems to be a preliminary work that needs deeper study on this topic, for example, how to efficiently (with less compute/less data points) align model with human value? Does different value affect each other during post-training? Can one alter the model output’s value by performing another round of SFT/RL, if so, how to achieve it in a more effective way?
2. It would be interesting to check if any of the dataset used in the SFT and RL stages actually contain data points that are related to the topic/values examine in the evaluation phase. If they do contain data points, say related to immigration, then does any of the data points indeed represent a negative/neutral/positive viewpoint which later on pick-up by the model? This is not clear to me. I think the proportion of related data in the training datasets are more important than algorithm used for the post-training phases, however, this is not examined thoroughly by the authors.
3. The coverage of datasets/model/post-training algorithm is not comprehensive, which make the conclusion/observations in the paper not very convincing. Even though there are compute constraints, there should be explanation on why only examining two model family and small size model on limited datasets will be enough to draw the conclusion.

**Questions:**

1. Can someone just inject data points with strong opinion in the training dataset used by SFT or RL, and completely alter a model’s value? Is it more effective to do so during post-training than pre-training? What about PEFT algorithm for fine-tuning? This is not studied in the paper.

---

> ### Author Response · Authors · 2025-11-26
> **Response to Reviewer L3se (1/3)**
>
> We thank the reviewer for noticing our work as an interesting and important topic and appreciate the positive comments regarding the clarity of the paper. We address the reviewer’s questions below:
>
> **W1:** *Overall, the paper is more like an empirical study of model value alignment. It did not propose novel algorithm or framework to perform better model human value alignment [...] achieve it in a more effective way?*
>
> We agree that our work is an empirical investigation of value alignment dynamics during LLM post-training, and we do not claim otherwise. **However, we respectfully disagree that the contribution is limited.**
>
> Our **primary contribution is to perform a systematic analysis of** how and when models acquire values throughout LLM post-training, which, to our knowledge, has not been previously studied at this scale and level of granularity. Our study provides a comprehensive analysis of tracing value alignment and novel insights which we believe are a significant contribution. Below, we summarize our main insights which are both novel and practically significant:
>
> 1. Performing preference optimization on commonly used preference datasets results in minimal to no value drift, which means that a model does not acquire values in the preference optimization phase, regardless of the chosen algorithm.
> 2. We show that the reason for the minimal value drift is that existing datasets lack a meaningful value gap, as chosen and rejected preferences are often stylistic rather than value-driven.
> 3. We do create a new dataset, i.e., a synthetic preference dataset with a controlled value gap, and show that different preference optimization algorithms can produce different value alignment outcomes, yet differ in how fast values drift during post-training
>
> These results provide the **first systematic analysis** of when and how model values evolve during post-training, offering actionable insights for designing post-training pipelines, from dataset curation to algorithm selection.
>
> We acknowledge that we do not propose a new algorithm or benchmark. However, **we believe that this should not be considered a weakness** as many prior ICLR papers [1,2,3] have made valuable contributions by conducting careful empirical studies to better understand specific phenomena.
>
> We appreciate the reviewer’s suggestions regarding efficiency (e.g., aligning with fewer data points or less compute) and iterative modification of values through additional rounds of SFT/RL. Though these are interesting research questions, they are outside the scope of our work, which focuses on analyzing the default post-training pipeline that serves as the backbone for major foundational models like Qwen2 [4], Olmo [5], and LLaMA [6] (as described in Sections 2.2 and 2.3), but remain valuable directions for future research.
>
> ### **Does different value affect each other during post-training?**
>
> Thank you for this interesting question. To examine whether values affect each other during post-training, we run a targeted experiment using the synthetic support-aligned dataset (see Appendix M in revised manuscript) and perturb only one topic, Immigration while keeping all others fixed. We flip Immigration’s chosen preference in two settings (1) either 30% support / 70% oppose (so opposing immigration is the majority chosen preference) or (2) 70% support / 30% oppose (so supporting immigration is the majority chosen preference). This lets us test how altering the preference signal for one value affects value drifts for the same value and across other values.
>
> We use Llama-3-3B (SFT-initialized with WildChat) for our experiments. We find that PPO and SimPO show minimal to no value drifts under both perturbation settings. However, when using DPO, for the first perturbation setting, we observe modest value drifts as support remains the majority chosen preference in the dataset, so the policy is not able to amplify the opposing value for immigration (see Fig. 19a), despite immigration having a majority opposing stance as chosen preference. On the other hand, for the second perturbation setting, support is the majority preference for all values, including immigration, and the policy under DPO is able to amplify this preference strongly (see Fig. 19b). Finally, PPO and SimPO showed no significant cross-value effects, whereas DPO exhibited minimal to modest drift.

---

> ### Author Response · Authors · 2025-11-26
> **Response to Reviewer L3se (2/3)**
>
> **W2:** *It would be interesting to check if any of the dataset used in the SFT and RL stages actually contain data points that are related to the topic/values [...] however, this is not examined thoroughly by the authors.*
>
> **Response:** We thank the reviewer for this question. We agree that disentangling the dataset distributions is an important aspect. However, a full audit of the datasets is beyond the scope of this work. Instead, we are only able to approximate the underlying stance distribution of the datasets through two methods.
>
> *Approach 1* - Embedding-based retrieval: We embed each SFT dataset (WildChat and Alpaca) and all value-probing questions in V-PRISM using the all-mpnet-base-v2 sentence transformer. For every question in V-PRISM, we compute cosine similarity against all dataset embeddings and retain datapoints with similarity ≥ 0.5. We then classify the stance for each retrieved datapoint, following the same evaluation protocol used for model outputs (§3). Using this method, we retrieve 5996 datapoints from WildChat (≈2.3%) and 612 datapoints from Alpaca (≈1.3%).
>
> *Approach 2* - Keyword-based retrieval: To capture a broader coverage, we use a keyword-based search aligned with the 11 topics in V-PRISM. This way, we retrieve 37856 datapoints from WildChat dataset (≈14.8%) and 2110 datapoints from Alpaca dataset (≈4.5%), and apply our stance classification pipeline to these datapoints.
>
> We then plot these stance distributions, as shown in Appendix K. We observe that WildChat exhibits a predominantly neutral stance, with 72.3% of its retrieved data points classified as neutral. On the other hand, Alpaca exhibits a clear supportive stance, with a majority (67%) of its datapoints classified as supportive across all topics.
>
> Similarly, for standard preference datasets, we retrieve 10177 datapoints from UltraFeedback (16.7%) and 7627 datapoints from HH-RLHF (12.5%). We then calculate and plot the Euclidean distances between the stance distribution of the ‘chosen (preferred)’ and ‘rejected (dispreferred)’ responses. As shown in Figure X, we observe that both datasets are highly concentrated at the low end of the distance scale, exhibiting a large peak near 0. This indicates a low value gap and weak signal for preference optimization.
>
> In contrast, for our synthetic dataset with a controlled value gap, the Euclidean distances show a dominant peak at the high-distance end, demonstrating a substantial number of preference pairs with a strong value difference, providing a clearer signal for preference learning.
>
> ---
>
> **W3:** *The coverage of datasets/model/post-training algorithm is not comprehensive, which make the conclusion/observations in the paper not very convincing. Even though there are compute constraints, there should be explanation on why only examining two model family and small size model on limited datasets will be enough to draw the conclusion.*
>
> We are surprised that the reviewer found our coverage insufficient. We conduct numerous experiments across multiple models, datasets, and post-training algorithms to reach our conclusions. Below, we summarize the scope of our experiments:
>
> - **Base models:** LLaMA3-3B, LLaMA3-8B, Qwen3-4B, Qwen3-8B
> - **SFT models:** Trained on WildChat and Alpaca, 11 checkpoints each
> - **Preference optimization:** PPO, DPO, SimPO using HH-RLHF and UltraFeedback datasets (48 models, evaluated 384 checkpoints)
> - **Alignment scenarios:** Support-aligned and oppose-aligned datasets (48 models, evaluated 384 checkpoints)
> - **Checkpoints evaluated:** 860 checkpoints across 104 trained models
> - **Evaluation prompts:** 550 prompts from V-PRISM, 5 generations each → 2,365,000 total generations (over 2M)
> - **Cost:** Total cost of OpenAI API ~$1,200
>
> We believe that these experiments provide comprehensive coverage across model families, dataset types, and algorithms. This ensures that our conclusions are well-supported and statistically meaningful. We also like to highlight that our framework is model-agnostic and can be applied to study any algorithm or model in future work.

---

> ### Author Response · Authors · 2025-11-26
> **Response to Reviewer L3se (3/3)**
>
> **Q1:** *Can someone just inject data points with strong opinion in the training dataset used by SFT or RL, and completely alter a model’s value? Is it more effective to do so during post-training than pre-training?*
>
> **Response:** This is an interesting question. We use our experiment from W1 to answer this question (also detailed in Appendix M in the revised manuscript). We run a targeted experiment using the synthetic support-aligned dataset and perturb only one topic, Immigration, while keeping all others fixed. We flip Immigration’s chosen preference in two setups:
>
> (1) either 30% support / 70% oppose (so opposing immigration is the majority chosen preference) or
>
> (2) 70% support / 30% oppose (so supporting immigration is the majority chosen preference).
>
> This lets us test how altering the preference signal for one value affects value drifts for the same value and across other values.
> In Setup 1, we observe that DPO tries to move the policy towards the majority preference in the dataset, but with less drift magnitude since opposition is the dominant chosen preference for immigration. Also, we observe cross-value influence for topics like Race and Racism, with the neutral stance ending up as the dominant response (see Fig. 19a and Fig. 20a).
>
> For Setup 2, we observe that DPO amplifies the majority chosen preference (support for all topics, including immigration), reinforcing support as the dominant stance for Immigration. The effect on all topics remains the same, including for Race and Racism, where we observe that DPO amplifies support and becomes the dominant stance (see Fig. 19b and Fig. 20b).
>
> With PPO and SIMPO, we observe minimal to value drifts regardless of the setup, which shows that there is less influence to alter a model’s acquired values after SFT, even if one tries to inject datapoints with strong opinions.
>
> ---
>
> **Q2.**  *What about PEFT algorithm for fine-tuning? This is not studied in the paper.*
>
> **Response:** To isolate the effects of standard fine-tuning and alignment objectives on the model’s entire parameter space, our main experiments focus on full fine-tuning (updating all parameters). However, recognizing that many practical setups use parameter-efficient fine-tuning methods, we conduct an additional experiment using LoRA to study how value alignment behaves under PEFT. We follow the training setup by [7], and train the Llama3-3B model using the Alpaca dataset for SFT and use our support-aligned dataset for preference optimization. We report full results in Appendix N in the revised manuscript. Across all settings, we observe the same model behavior as in the full fine-tuning regime.
>
> ### **References**
>
> [1] Kirk et al. Understanding the Effects of RLHF on LLM Generalisation and Diversity. ICLR 2023
>
> [2] Sharma et al. Towards Understanding Sycophany in Language Models. ICLR 2024
>
> [3] Ye et al. Justice or Prejudice? Quantifying Biases in llm-as-a-judge. ICLR 2025
>
> [4] Qwen Team. Qwen2 Technical Report
>
> [5] Olmo Team. 2 OLMo 2 Furious
>
> [6] LLaMA Team. The Llama 3 Herd of Models
>
> [7] Thakkar et al., A Deep Dive into the Trade-Offs of Parameter-Efficient Preference Alignment Techniques. ACL 2024

---

### Official Review · Reviewer_gcnn · 2025-10-31

**Soundness:** 2
**Presentation:** 2
**Contribution:** 3
**Rating:** 2
**Confidence:** 3

**Summary:**

The work investigates how at each stage the value of the alignment drifts during the model's post-training. The findings indicate that different preference optimization algorithms lead to different value outcomes. To measure these drifts, the work construct V-PRISM consists of a total of 550 prompts. The drifts are then calculated by classifying the stance of the model at regular timestamps. The model's post-training drifting is measured using two metrics: drift magnitude and drift time.

**Strengths:**

- The study evaluated several post-training methods, including preference optimization, to measure the value drifts.
- The paper explores whether the low value gap in standard preference datasets results in low-value drift by using two distinct scenarios of support-aligned and oppose-aligned, where the preferred labels are switched.

**Weaknesses:**

- There is no explanation provided for why models adhere to the values that were aligned during the SFT phase of preference optimization. All results are discussed empirically, which are also limited. Additionally, there is limited discussion on how different datasets used during SFT result in varying magnitudes of drift during the preference optimization.
- The study fails to offer practical recommendations for mitigating value drifts during preference optimization for algorithms such as DPO. This results in increased drifts compared to PPO and SimPO, or when an oppositely aligned dataset is employed to maliciously manipulate the alignment of LLMs.
- I am not sure about the reasoning behind studying the drift for PPO, as it has a KL-divergence term in its formulation to mitigate the same issue. The paper also mentions that PPO has the lowest drift with this explanation. The contributions would be much higher if other algorithms like DAPO had been studied that do not include the KL divergence terms.
- The stance is only limited to 3 options. It does not tell how much the model is in support or opposition to a topic. Furthermore, measuring the stance of a language model using an LLM (GPT-4o) is not a robust way to do so, as the LLM will have its biases. A better approach would be to get stances from multiple LLMs or incorporate human evaluation.
- The rationale behind employing the oppose-aligned dataset remains unclear. Preference optimization aims to align the model’s behavior with desired preferences. Consequently, utilizing the opposing-aligned data to widen the gap will undoubtedly lead to a shift in value.

**Questions:**

What is the reason behind selecting Llama3 and Qwen3? Why specifically choose Llama3 and not the other variants?

---

> ### Author Response · Authors · 2025-11-26
> **Response to Reviewer gcnn (1/3)**
>
> We thank the reviewer for providing constructive feedback. We address the reviewer’s questions below.
>
> **W1:**  *There is no explanation provided for why models adhere to the values that were aligned during the SFT phase of preference optimization. All results are discussed empirically, which are also limited. Additionally, there is limited discussion on how different datasets used during SFT result in varying magnitudes of drift during the preference optimization.*
>
> **Response:**  As discussed in Section 2.2, the goal of SFT is to maximize the log-likelihood of responses given instructions, teaching the model to follow instructions and mimic the data distribution. Importantly, SFT is **not performed with an explicit objective of value alignment**, yet our findings show that even general-purpose instruction-following datasets like WildChat and Alpaca impact different value profiles in the model, **even though neither is designed explicitly for value learning.**
>
> Regarding the comment on dataset effects, **we respectfully disagree that this was not addressed**. We explicitly evaluate models trained with different SFT datasets (WildChat and Alpaca) and examine how subsequent preference optimization behaves for each. Our findings (as shown in Section 6) in fact demonstrate that the choice of SFT dataset influences the drift magnitude and drift time, illustrating that both the dataset and the optimization algorithm jointly shape alignment outcomes. For example, as illustrated in Fig. 4a, 4b, 4c, SFT models trained with WildChat have a higher probability of neutral stance, whereas SFT models trained with Alpaca have a higher support stance, and when the preference signal aligns with this SFT prior, DPO (Fig. 4b) can amplify this stance.
>
> ---
>
> **W2:** *The study fails to offer practical recommendations for mitigating value drifts during preference optimization for algorithms such as DPO. This results in increased drifts compared to PPO and SimPO, or when an oppositely aligned dataset is employed to maliciously manipulate the alignment of LLMs.*
>
> **Response:** Thank you for this comment. We would like to clarify that our goal is to ***characterize*** rather than ***mitigate*** value drift, as such shifts are not inherently detrimental and may be necessary for adapting models to evolving requirements. Our objective is to trace how and when models acquire values and disentangle the effects of datasets versus algorithms during LLM post-training. In Section 5, we show that when using standard preference datasets and varying the algorithm used for preference optimization, all three widely used algorithms (PPO/DPO/SIMPO) exhibit **minimal to no value drifts**. In Section 6, we introduce a controlled value gap into the preference dataset and examine how the same algorithms lead to different value alignment outcomes. The use of the oppose-aligned dataset is not to demonstrate malicious manipulation, but for a **controlled experiment** to investigate the alignment algorithm behaviour when the preference gap is maximum, and study how values drift in such settings where there is an explicit signal in the preference dataset. In our experiments, we see that our SFT models have a low probability of oppose stance (see Appendix K.1 in the revised manuscript); therefore, the chosen preference signal (oppose) explicitly conflicts with the SFT prior in most cases. Our findings do yield practical insights for practitioners, such as how dataset design (e.g., value gaps) and the choice of algorithm jointly influence value alignment outcomes. Overall, our work aims to characterize when and why value drifts arise, not to mitigate them.

---

> ### Author Response · Authors · 2025-11-26
> **Response to Reviewer gcnn (2/3)**
>
> **W3:** *I am not sure about the reasoning behind studying the drift for PPO, as it has a KL-divergence term in its formulation to mitigate the same issue. The paper also mentions that PPO has the lowest drift with this explanation. The contributions would be much higher if other algorithms like DAPO had been studied that do not include the KL divergence terms.*
>
> **Response:** We include PPO because it is the earliest and most widely adopted preference-optimization algorithm, and many subsequent preference optimization variants build directly on it. As noted by the reviewer, PPO’s explicit KL-divergence penalty constrains deviations from the SFT reference model. Our findings reveal that when PPO is used as the preference optimization algorithm, the model largely preserves the values acquired during SFT, leading to minimal drift. More broadly, **our study is not aimed at mitigating drift as highlighted in W2 but at understanding how and when value changes arise in the post-training pipeline** and how both datasets and algorithms shape these outcomes. In this context, PPO provides an important baseline, demonstrating that when the KL penalty is high, the model will not learn any new values during the preference optimization phase, regardless of the preference data used (whether one uses standard preference datasets as discussed in Section 5, or uses synthetic preference data with controlled value gap as shown in Section 6).
>
> We appreciate the reviewer’s suggestion to include DAPO. While DAPO is a promising algorithm for reasoning and verifiable domains (e.g., Math and Coding tasks), it is not currently suitable for value alignment tasks for the following reasons:
>
> 1. **Incompatibility of Reward Signals:** DAPO relies on “Dynamic Sampling,” which assumes an objective ground truth to filter prompts based on binary correctness. However, value alignment lacks this ground truth, and instead our study relies on continuous, subjective preference scores from a reward model. This means that we cannot directly apply DAPO's sampling technique without significant modification.
> 2. **Scope of Contribution:** Adapting DAPO to value alignment would require designing entirely new sampling heuristics for subjective rewards, which constitutes a separate research contribution beyond the scope of this paper and we motivate this for future research.
>
> However, to address the underlying interest in KL constraints, we note that we have explicitly studied the effect of varying KL divergence coefficients on PPO performance in Appendix J.1.
>
> ---
>
> **W4:** *The stance is only limited to 3 options. It does not tell how much the model is in support or opposition to a topic. Furthermore, measuring the stance of a language model using an LLM (GPT-4o) is not a robust way to do so, as the LLM will have its biases. A better approach would be to get stances from multiple LLMs or incorporate human evaluation.*
>
> **Response:** We limit the stance space to three categories (support, neutral, oppose) to keep the analysis tractable and aligned with prior stance-detection work that also uses three-way labels [1, 2, 3]. We initially experimented with more fine-grained options (e.g., mild support / mild oppose), but found that these categories had consistently negligible probability mass across topics, which made the analysis noisier without providing additional insight. Qualitative analysis also showed that such subtle stances were subjective and difficult to quantify reliably.
>
> Regarding the use of GPT-4o for stance extraction, this task is relatively straightforward. As noted in L189–L191, one author manually validated a random sample of 100 prompt–generation pairs and confirmed that GPT-4o’s classifications were consistent with human judgment (agreement score 92%). To further assess the robustness of our evaluation, we also compute stance distributions using Gemini 2.5 Pro for 25,000 prompt-generation pairs. Across models, we observed high label agreement (93.38%), very small distributional differences (mean JSD = 0.07), and strong correlation (r=0.89) between predicted probabilities. These results indicate that our conclusions are not sensitive to the choice of LLM used for stance measurement.

---

> ### Author Response · Authors · 2025-11-26
> **Response to Reviewer gcnn (3/3)**
>
> **W5:** *The rationale behind employing the oppose-aligned dataset remains unclear. Preference optimization aims to align the model’s behavior with desired preferences. Consequently, utilizing the opposing-aligned data to widen the gap will undoubtedly lead to a shift in value.*
>
> **Response:** The rationale for using the oppose-aligned dataset is to maximize the preference gap and test the model's ability to learn this explicit signal in the preference dataset. In this setting, the chosen preference always corresponds to opposing the topic, while the rejected preference corresponds to always supporting the topic. For example, the chosen preference is opposing immigration and climate change, and the rejected preference is supporting immigration and climate change. First of all, your understanding of preference optimization is correct: the algorithm should, in principle, align the model toward the chosen preference (i.e., consistently opposing each topic). However, our results show that this does not happen in practice, regardless of the preference optimization method used.
>
> As shown in Figure 4, neither PPO nor SIMPO meaningfully shifts the model toward the chosen stance (oppose). In Figure 4a (PPO), the stance probabilities remain unchanged from the SFT (see the red curve), indicating that PPO fails to adopt the chosen preference. Figure 4c (SIMPO) shows a similar pattern. Only DPO (Figure 4b) increases the probability of the chosen stance, but even then, the change is limited, since the oppose stance has a very low SFT prior, so the policy only moves partway and never adopts the oppose-value as the dominant stance. So, for DPO training, the drift is large and fast (high drift magnitude and low drift time) only when the SFT model already assigns a relatively high probability to the chosen preference (as shown in the support-aligned condition in Figure 4b).
>
> Therefore, contrary to the reviewer’s assertion, using oppose-aligned data does not “undoubtedly” produce a value shift. In fact, our paper reveals cases where preference optimization fails to shift the model toward the chosen preference at all.
>
> ---
>
> **Q1:** *What is the reason behind selecting Llama3 and Qwen3? Why specifically choose Llama3 and not the other variants?*
>
> **Response:** We select Llama3 and Qwen3 models since they are the most widely used and actively maintained open-weight model families. The broad adoption of these models in both research and industry ensures that our findings are relevant to current practices. In addition, both families provide multiple model sizes, which allows us to examine value alignment dynamics across scales. In our experiments, we use smaller (3B–4B) and larger (8B) variants to test whether our findings are consistent across model scales. Overall, we believe these reasons make Llama 3 and Qwen3 well-suited for our experiments.  We also want to highlight that this focused model selection enabled a comprehensive evaluation, covering **104 models, 860 checkpoints, and a total of 2,365,000 generations**. This scale ensures that our observations regarding value alignment dynamics are consistent and statistically significant across models and training stages.
>
> ## References
>
> [1] Xu et al., OpenStance: Real-world Zero-shot Stance Detection. CoNLL 2022
>
> [2] Taranukhin et al., Stance Reasoner: Zero-Shot Stance Detection on Social Media with Explicit Reasoning. COLING 2024
>
> [3] Kawintiranon et al., Knowledge Enhanced Masked Language Model for Stance Detection. NAACL 2021

---

> > ### Comment · Reviewer_gcnn · 2025-11-27
> >
> > Thank you for the detailed response and adding additional material in the appendix for clarity. I have updated the score.
> >
> > Thank you for the detailed response and for adding additional material in the appendix for clarity. I have updated my score accordingly.
> >
> > I understand that the paper provides a systematic analysis of when and how model values evolve during post-training. However, the contributions of the paper are still limited. The authors state that the paper is not about mitigating value drift or presenting a novel algorithm to efficiently align the model based on the analysis, but the study nevertheless fails to provide clear, actionable takeaways regarding which preference algorithms should be used in which scenarios. While the paper includes many experiments, it lacks explanations for why particular algorithms perform better or worse across different models and datasets. The study would be significantly stronger if the results were accompanied by principled explanations of why each preference algorithm behaves as it does, rather than only presenting what the experiments show.

---

### Official Review · Reviewer_kSKx · 2025-11-01

**Soundness:** 2
**Presentation:** 3
**Contribution:** 2
**Rating:** 2
**Confidence:** 4

**Summary:**

The paper analyzes value drifts in LLMs during training, including SFT and preference learning. The paper constructs a set of value-laden prompts and an LLM-as-judge to assess the stance of the model responses across training. The paper finds for its setup (i.e., the specific datasets considered in each stage of training) that for the most part value distributions change the most at the beginning of SFT and minimally thereafter, except when training with DPO on preference data where the responses are forced to represent different stances.

**Strengths:**

1. The paper tackles an important problem (value alignment) from a novel / underexplored perspective (training dynamics).
2. The paper is well-written, with clear figures, tables, experimental setups, etc.
3. The paper showcases differences in values alignment throughout training between preference learning algorithms which could be interesting avenue of further study.

**Weaknesses:**

The main missing piece to this analysis in my mind are understanding the datasets themselves:
1. To be able to say anything meaningful about the claim that models learn during SFT, it seems important to disentangle the stance distribution of the SFT datasets in the analysis. For instance, how closely do the models match the distributions of the datasets used?
2. The SFT vs. preference learning comparison does not disentangle the fact that the datasets are using different query distributions. This seems like a very important aspect to the analysis to consider. For instance, what happens when the datasets used for SFT vs. preference learning are switched? (This general question could be asked with controlled testing using synthetic data).

**Questions:**

See the points under weaknesses above.

Additional question: Can the authors provide some additional analysis / explanation of the differences between DPO and SimPO seen in Figure 4?

---

> ### Author Response · Authors · 2025-11-26
> **Response to Reviewer kSKx (1/2)**
>
> We thank the reviewer for noticing our work as a novel and underexplored perspective. Below, we address the reviewer's concerns
>
> **W1:** *To be able to say anything meaningful about the claim that models learn during SFT, it seems important to disentangle the stance distribution of the SFT datasets in the analysis. For instance, [...] datasets used?*
>
> **Response:** We agree that disentangling the dataset distributions is an important aspect, however a full audit of the dataset is beyond the scope of this work. Instead, we are only able to approximate the underlying stance distribution of SFT datasets through two methods.
>
> *Approach 1:* Embedding-based retrieval. We embed each SFT dataset (WildChat and Alpaca) and all value-probing questions in V-PRISM using the all-mpnet-base-v2 sentence transformer. For every question in V-PRISM, we compute cosine similarity against all dataset embeddings and retain datapoints with similarity ≥ 0.5. We then classify the stance for each retrieved datapoint, following the same evaluation protocol used for model outputs (§3). Using this method, we retrieve 5996 datapoints from WildChat (≈2.3%) and 612 datapoints from Alpaca (≈1.3%).
>
> *Approach 2:* Keyword-based retrieval. To capture a broader sense of the data distribution, we use a keyword-based search aligned with the 11 topics in V-PRISM. This way, we retrieve 37856 datapoints from WildChat dataset (≈14.8%) and 2110 datapoints from Alpaca dataset (≈4.5%), and apply our stance classification pipeline to these datapoints.
>
> We then plot these stance distributions, as shown in Appendix K1 in the revised manuscript. We observe that WildChat exhibits a predominantly neutral stance, with 72.3% of its retrieved data points classified as neutral. On the other hand, Alpaca exhibits a clear supportive stance, with a majority (67%) of its datapoints classified as supportive across all topics.
>
> To measure how closely SFT-trained models reflect the stance distribution of the underlying data, we calculate the Wasserstein distance of stance distributions between (1) SFT data and SFT model, (2) SFT data and Base model, and (3) SFT model and Base model. First, we compute the correlation between (2) and (3) and find high to moderate correlations across all topics [0.4981–0.8387]. This shows that as the Wasserstein distance between SFT data and base model increases, the distance between the SFT model and base model also increases. To further confirm that the SFT model distribution is mimicking the SFT data distribution, we compare the mean Wasserstein distance in (1) and (2) and find that the mean distance is much lower in (1) as compared to (2). These two observations confirm that SFT is closely mimicking the stance distribution of the underlying SFT data.
>
> We want to emphasize that it’s not particularly surprising that SFT is mimicking the data distribution because that’s precisely the training objective. **However, it is surprising that later stages of preference optimization are largely unable to overwrite the values induced by SFT.** This provides novel insights into the role of the SFT dataset for value alignment and has clear practical implications for future work on post-training.
>
> ---
>
> **W2.** *The SFT vs. preference learning comparison does not disentangle the fact that the datasets are using different query distributions. This seems like a very important aspect to the analysis to consider. For instance, [...] are switched?*
>
> **Response:** Yes, the query distributions of SFT and preference datasets differ. To isolate the effect of query distribution from the effect of value-gap, we conduct additional experiments that explicitly switch the datasets.
>
>
> (1) **SFT using the preference dataset.** We take the chosen responses from UltraFeedback and HHRLHF datasets and perform SFT using Llama3-3B as the base model. We evaluate all checkpoints using our standard protocol, and the results (Appendix L1 in the revised manuscript) show consistent behaviour i.e., SFT strongly and rapidly initializes the model’s values and different SFT datasets impart different value profiles.
>
> (2) **Preference optimization using SFT dataset.** Using our embedding-based retrieval method, we extract 6608 topic-relevant prompts from WildChat and Alpaca. We then construct a synthetic preference dataset with controlled value-gaps following the support-aligned setup, as described in Section 6.2. We run PPO, DPO, and SIMPO preference optimization algorithms using these datasets and report results for three topics in Appendix L2 in the revised manuscript.
>
> Across both experiments, the outcome is consistent: **switching the query distributions does not change the conclusions.** SFT is still the dominant driver of value alignment and different preference optimization algorithms produce the same value alignment outcomes when we control the value-gap between chosen and rejected responses.

---

> ### Author Response · Authors · 2025-11-26
> **Response to Reviewer kSKx (2/2)**
>
> **Q1:** *Can the authors provide some additional analysis / explanation of the differences between DPO and SimPO seen in Figure 4?*
>
> **Response:** As discussed in Section 2.3, the differences between DPO and SimPO arise from how each method interprets preference data and defines the underlying reward signal.
>
> DPO reparameterizes the reward function in PPO to directly learn a policy model from the preference data, eliminating the need for an explicit reward model. The implicit reward is formulated using the log ratio of the likelihood of a response between the current policy model and the SFT (reference) model. This means that though the policy is encouraged to increase the relative likelihood of the chosen preference, it remains anchored to the reference. We see this in our findings, as shown in Figure 4 and Section 6.2. When the chosen preference aligns with the SFT (reference) model, the policy under DPO is able to amplify this. However, when the chosen preference misaligns with the SFT model, the policy shifts partway towards the chosen preference and is not adopted as the dominant stance.
>
> SimPO, in contrast, removes the reference model entirely and instead relies on a target reward margin, which enforces that the likelihood of preferred (chosen) response must exceed the likelihood of non-preferred (rejected) response by at least a fixed margin. Once this margin is satisfied, the policy update becomes minimal. As a result, SimPO leads to modest value drifts, as illustrated in Figure 4. We also note that we have explicitly studied the effect of varying the target margin parameter in SimPO in Appendix J.3.

---

### Author Response · Authors · 2025-11-27
**General Response for all Reviewers**

We thank all reviewers for their thoughtful and constructive feedback. We are grateful that the reviewers recognised the **importance of the problem setting** and the **novelty of our contribution**. We also appreciate the positive remarks regarding the **clarity, level of detail, and potential usefulness of our analysis** for future research in this area.

We are committed to addressing all concerns raised by the reviewers and have substantially expanded and updated the manuscript in response (refer to *“Additional Experiments for Rebuttal”* section in blue font after the appendix). Below, we summarize the major additions:
1. **Stance distributions of all training datasets:** We now approximate and report stance distributions for all SFT and preference datasets, allowing clearer interpretation of dataset-induced priors (Appendix K).
2. **Switching query distributions:** We perform controlled experiments in which SFT and preference datasets are swapped. The outcomes remain consistent with our original conclusions. (Appendix L).
3. **Injecting strongly opinionated datapoints:** We analyze preference optimization when datasets are explicitly seeded with strong opinions, providing additional evidence on how values affect each other. (Appendix M).
4. **PEFT-based fine-tuning:** We repeat key experiments using LoRA to examine whether parameter-efficient fine-tuning alters value drift dynamics (Appendix N).
5. **Bimodal and skewed preference distributions:** We study preference datasets with varied stance distributions to evaluate how distributional shape affects alignment outcomes (Appendix O).

We believe these additions significantly strengthen the manuscript by deepening the empirical analysis, clarifying the role of datasets, and expanding the generality of our findings. We welcome any further questions and are happy to provide additional clarifications.

---

### Meta-Review · Area_Chair_nGBQ · 2026-01-05

**Summary:**

This paper focuses on LLM value alignment during post-training with a specific aim of disentangling the effects of post-training algorithms and datasets, measuring both the magnitude and time of value drifts during training. The authors trace value drift across multiple models, datasets and post-training algorithms and find that different preference optimization algorithms produce different value alignment results, even with the same preference data.

**Reviewer Concerns:**

Reviewer concerns include:
Even after the authors addressed the concerns of the reviewers, substantial concerns remain. These include:

* Lack of novelty as the work is more “confirming of SFT” rather than providing substantial new contribution

* Lack of deeper theoretical insight made the contribution of the paper somewhat linked. Explanations for why particular algorithms perform a certain way and some actionable takeaways would have remedied this.

* Data analysis is partial and did not fully-explore the underlying distributions of SFT or preference datasets. Although this was partially addressed in the rebuttal via embedding/keyword-based retrieval, this only partially mitigates the concern.

**Reviewer Scores:**

The paper received Rating 2 (reject) from three reviewers while teh fourth one gave it a Rating of 4 (marginally below the acceptance threshold. But would not mind if paper is accepted). No rating calibrations were made following rebuttals.

---

### Decision · Program_Chairs · 2026-01-26

Reject